# Plk4 Is a Novel Substrate of Protein Phosphatase 5

**DOI:** 10.3390/ijms24032033

**Published:** 2023-01-19

**Authors:** Edit Ábrahám, Zsuzsánna Réthi-Nagy, Péter Vilmos, Rita Sinka, Zoltán Lipinszki

**Affiliations:** 1Biological Research Centre, Institute of Biochemistry, MTA SZBK Lendület Laboratory of Cell Cycle Regulation, ELKH, H-6726 Szeged, Hungary; 2Doctoral School of Biology, Faculty of Science and Informatics, University of Szeged, H-6726 Szeged, Hungary; 3Biological Research Centre, Institute of Genetics, ELKH, H-6726 Szeged, Hungary; 4Department of Genetics, University of Szeged, H-6726 Szeged, Hungary

**Keywords:** protein phosphatase 5, Polo-like kinase 4, centrosome, *Drosophila*, cell division

## Abstract

The conserved Ser/Thr protein phosphatase 5 (PP5) is involved in the regulation of key cellular processes, including DNA damage repair and cell division in eukaryotes. As a co-chaperone of Hsp90, PP5 has been shown to modulate the maturation and activity of numerous oncogenic kinases. Here, we identify a novel substrate of PP5, the Polo-like kinase 4 (Plk4), which is the master regulator of centriole duplication in animal cells. We show that PP5 specifically interacts with Plk4, and is able to dephosphorylate the kinase in vitro and in vivo, which affects the interaction of Plk4 with its partner proteins. In addition, we provide evidence that PP5 and Plk4 co-localize to the centrosomes in *Drosophila* embryos and cultured cells. We demonstrate that PP5 is not essential; the null mutant flies are viable without a severe mitotic phenotype; however, its loss significantly reduces the fertility of the animals. Our results suggest that PP5 is a novel regulator of the Plk4 kinase in *Drosophila*.

## 1. Introduction

The balance between phosphorylation and dephosphorylation is essential for the maintenance of cellular homeostasis in eukaryotes. Phosphorylation is regulated by the antagonistic activity of protein kinases and phosphatases in a highly complex and, usually, conserved manner. The orchestrated interplay between these two enzyme groups can fine-tune and regulate complicated cellular events, such as cell cycle progression. Although the role of kinases governing cell division is well established, we know much less about the function and substrates of the protein phosphatases.

The evolutionarily conserved protein phosphatase 5 (PP5) belongs to the family of Ser/Thr phosphoprotein phosphatases (PPP: includes PP1, PP2A, PP2B, PP4, PP6 and PP7) [1,2,3,4]. PP5 functions as a monomer and does not employ conventional, non-covalently bound regulatory subunits for substrate recognition or for control of its catalytic activity. All of these features are encoded by its primary amino acid sequence [5,6]. Three tetratricopeptide (TPR) motifs are located at the N-terminal region of PP5 (Appendix A), which fold back onto the C-terminal catalytic (phosphatase) domain, keeping the basal activity of PP5 low [7]. This auto-inhibition can be released by the binding of fatty acids or the Hsp90 chaperone to the TPR motifs, as well as by phosphorylation of the conserved Thr362 (in human PP5) [8,9,10,11]. It has been reported that the TPR motifs alone, as well as the catalytic domain itself, are capable of substrate binding [12]. 

PP5 regulates diverse cellular processes, including cell growth and differentiation [13], DNA damage checkpoints and repair [14,15,16], p53 activity [17], stress response [18], circadian rhythm [19] and receptor signaling [5]. Impaired function of PP5 has been associated with different diseases, such as diabetes, obesity and cancer, therefore PP5 has recently become a potential therapeutic target (reviewed in [6]). Its function in cell division has also been proposed, due to the presence of the TPR motifs. Ollendorff and Donoghue have reported that PP5 localizes to the mitotic spindle in Cos-1 mammalian cells via its TPR motifs, and physically interacts with subunits of the APC/C ubiquitin ligase required for proper metaphase/anaphase transition during mitosis [20]. The same research suggested centrosomal localization of PP5, which was also observed in hTERT-RPE1 mammalian cells [21]. RNAi screen in the Glover lab has revealed that the knock-down of *Drosophila pp5* (*PpD3*) creates a mild centrosome number phenotype [22]. These observations suggest that PP5 might also play a role in centrosome biogenesis that is governed by Polo-like kinase 4 (Plk4). Interestingly, in a high-throughput protein-trap screen, Plk4 was co-purified with the affinity-tagged PP5 in *Drosophila* embryos [23], which has never been further investigated. 

Plk4 is a conserved oncogenic kinase that is often referred to as the master regulator of centriole duplication [24,25,26,27]. Centrioles are the main structural components of centrosomes, which become the major microtubule organizing centers in mitotic cells in animals. Therefore, their duplication must be carried out in a semi-conservative manner with extremely high fidelity, once every cell cycle [28,29]. Not surprisingly, the misregulation of Plk4 leads to serious mitotic defects, including chromosome segregation and centrosome number alterations, common hallmarks of cancer cells. These changes can promote tumorigenesis, making Plk4 a highly studied target in cancer diagnostics and therapy (reviewed in [30]). Comprehensive studies have revealed that the expression, localization, activity and half-life of Plk4 are tightly regulated at pre- and post-transcriptional, as well as post-translational, levels (reviewed in [30]). The mature homo-dimeric Plk4 kinase is primarily regulated by self-phosphorylation and SCF-mediated ubiquitination in a complex manner. Phosphatases might also be involved in Plk4 regulation [31], however, it is unknown exactly how these enzymes (and which enzymes besides PP2A) fine-tune the function of Plk4. 

In this study, we demonstrate that protein phosphatase 5 interacts with and dephosphorylates Plk4 in *Drosophila melanogaster* and human cells. We show that PP5 co-localizes with Plk4 to the centrosomes in fruit fly embryos and cultured cells, and speculate that PP5 might be involved in the precise phospho-regulation of Plk4.

## 2. Results

### 2.1. PP5 Interacts with Plk4

To better understand the role of *Drosophila* PP5 in cell division, we aimed to identify potential mitotic interactors of the phosphatase. In the *Drosophila* Interactions Database (http://www.droidb.org (accessed on 29 August 2018)), we found several interesting candidates, including Plk4 kinase [23]. To validate the interaction between PP5 and Plk4, we transiently co-transfected *Drosophila* D.Mel-2 cultured cells with GFP-PP5 (wild type) and the non-degradable (ND) form of Plk4 (Flag-Plk4-ND). We used the non-degradable mutant to prevent rapid proteasomal degradation of the transgenic kinase [32,33]. We found that GFP-PP5 specifically pulled-down Flag-Plk4-ND from D.Mel-2 cells (Figure 1a). Similarly, we co-purified Flag-HsPlk4-ND with GFP-HsPP5 from HEK293 human cells, as well (Figure 1b). These experiments suggest a conserved interaction between PP5 and Plk4. To test the direct binding between the two enzymes, we performed a GST-IVTT pull-down assay with *Drosophila* recombinant PP5 and Plk4 [34]. Immobilized GST-PP5 was incubated with ^35^S-methionine-labelled Plk4 (hereafter ^35^S-Plk4) synthesized in an in vitro coupled transcription and translation reaction (IVTT). Interacting proteins were resolved by SDS-PAGE and analyzed using autoradiography, which showed that Plk4 directly and specifically binds to GST-PP5 (Figure 1c). We also demonstrated that binding is independent of PP5’s activity, because Plk4 binds both to the inactive (GST-PP5^H326N^) and hyperactive (GST-PP5^E97Q^) forms of PP5 (Figure 2a and Appendix A).

### 2.2. PP5 Binds to the Kinase and PB1 Domains of Plk4

We wanted to clarify which domains or motifs of the two proteins are involved in the physical interaction of PP5 and Plk4. PP5 has three N-terminal TPR motifs, followed by a C-terminal catalytic domain and an αJ motif (Figure 2a and Appendix A) [6]. The TPR motifs are necessary for the protein-protein interaction and, together with the αJ sequence, are involved in the auto-inhibition of the phosphatase. Plk4 has an N-terminal kinase domain (KD), required for catalytic activity, followed by a coiled-coil region (CC), a downstream regulatory element (DRE), an L1 linker, Polo-box 1 and 2 domains (PB1-2), an L2 linker and a PB3 (Figure 2b). PB1 and PB2 form the cryptic Polo-box (CPB) that is needed for the homodimerization of Plk4 and the creation of a platform for interactor or substrate binding. PB3, along with the CC, are necessary to relieve the auto-inhibition caused by the L1 linker region of Plk4. The DRE and L2 regulate the stability of the enzyme [30]. 

First, we tested the in vitro interaction between GST-tagged PP5, PP5^203−520aa^ (containing the catalytic domain only) or PP5^1−202aa^ (containing the TPR motifs only), respectively, and IVTT-synthesized ^35^S-Plk4. We observed that Plk4 binds more strongly to the catalytic domain of PP5 (PP5^203−520aa^) than to the full length PP5 or the TPR motif-containing fragment (PP5^1−202aa^) (Figure 2a). We therefore concluded that unlike most partners of PP5, Plk4 favors binding to the catalytic domain. Next, we tested which domains/motifs of Plk4 can bind to GST-PP5. When we used truncated forms of ^35^S-Plk4, we discovered that the presence of the KD and CPB (in particular PB1) are critical for the interaction (Figure 2b). Surprisingly, however, when we used individual domains of ^35^S-Plk4, the CPB (PB1 + PB2) alone was not sufficient for binding, while the KD, L1 + PB1 and PB1 interacted strongly and specifically with GST-PP5 (Figure 2b). These findings indicate that the kinase domain and/or PB1 domain of Plk4 facilitate the direct interaction between Plk4 and the catalytic domain of PP5. 

### 2.3. PP5 Dephosphorylates Plk4

Next, we aimed to test whether PP5 can dephosphorylate Plk4, in vitro. We could not generate a wild type recombinant Plk4, because it is extremely unstable in bacteria. Therefore, in all in vitro experiments, we produced and used the bacterially-expressed, maltose-binding domain (MBP)-fused active recombinant Plk4 (hereafter MBP-Plk4-TE), which is capable of auto-phosphorylation in bacteria and remains active and phosphorylated upon purification [32]. First, we treated the purified MBP-Plk4-TE with the non-specific *λ*-phosphatase (*λ*PPase), as well as with the purified wild type (GST-PP5) or inactive (GST-PP5^H326N^) recombinant PP5 (Appendix A) in an ATP-free system, and analyzed the phosphatases’ effect with an in vitro gel-shift assay. This showed that the auto-phosphorylated MBP-Plk4-TE (which runs higher in the gel due to decreased electrophoretic mobility (e.m.)) is efficiently dephosphorylated (runs lower in the gel due to increased e.m.) by *λ*PPase, as well as by the GST-PP5 enzyme (Figure 3a). In both cases, we could detect an explicit gel-shift on SDS-PAGE, indicating that MBP-Plk4-TE was indeed dephosphorylated. The inactive GST-PP5^H326N^ was not able to modify Plk4. When MBP-Plk4-TE was incubated with a smaller amount of active GST-PP5, we could observe the relatively slow, but clear dephosphorylation kinetics (Figure 3b).

To validate this effect more specifically, we co-expressed MBP-Plk4-TE or its kinase dead version (MBP-Plk4-TEKM [32]), with His_6_-tagged PP5 (wild type, active) or PP5^H326N^ (inactive, Appendix A), respectively, in *E. coli* cells. Bacteria were lysed by boiling in Laemmli sample buffer to stabilize the in vivo conditions. We found that the permanent dephosphorylation of MBP-Plk4-TE requires active His_6_-PP5, in vivo. In the presence of inactive phosphatase, MBP-Plk4-TE remained phosphorylated, which is manifested by a slower migration of Plk4 in the SDS-PAGE gel (Figure 3c). As expected, neither the active nor inactive phosphatase had any effect on the kinase dead Plk4’s electrophoretic mobility (Figure 3c). Thus, we concluded that Plk4 is a genuine substrate of PP5.

We attempted to test that PP5 dephosphorylates Plk4 in *Drosophila* cells. Therefore, we co-expressed GFP-tagged non-degradable Plk4 (GFP-Plk4-ND) or its kinase dead version (GFP-Plk4-NDKD), with Myc-tagged PP5^E97Q^ (hyperactive, Appendix A) or PP5^H326N^ (inactive), respectively, in D.Mel-2 cultured cells. We could not detect significant differences in the electrophoretic mobility of GFP-Plk4-ND (co-expressed with either Myc-PP5^E97Q^ or PP5^H326N^). However, in the presence of inactive phosphatase, GFP-Plk4-ND appeared as a fuzzy and smeary band, which often indicates phosphorylation (Appendix A). This puts forward the possibility that GFP-Plk4-ND remained phosphorylated in the presence of the inactive PP5^H326N^, in the complex environment of wild type *Drosophila* cells. 

### 2.4. The Catalytic Activity of Plk4 Is Independent of PP5

It was reasonable to hypothesize that PP5 influences the activity, function, stability or binding-partner interaction (localization) of Plk4. We investigated this with in vitro assays. We have previously shown that in fruit flies, Plk4 phosphorylates the core centriole protein, Ana2, in a two-step reaction: first, Ser38 in the N-terminal ANST motif, then four serine residues in the C-terminal STAN motif [32,35]. In both cases, the phosphorylation-caused effect is detectable in an in vitro kinase assay, which is an indicator of functional Plk4. To test if Plk4’s kinase activity is altered by PP5-mediated dephosphorylation, MBP-Plk4-TE was expressed alone or co-expressed with His_6_-tagged active or inactive PP5, respectively, in *E. coli*. Then we tested whether the affinity purified kinases can phosphorylate Ser38 in the ANST motif of GST-Ana2^1−140aa^, as described earlier [35]. We found that neither the active PP5 (although it efficiently dephosphorylates the kinase, see Figure 3c) nor the inactive PP5 caused substantial decrease in Plk4’s ANST-motif phosphorylating activity. In all cases, Plk4 was able to phosphorylate Ser38 in GST-Ana2^1−140aa^, which manifested as a single discrete band-shift (Figure 4a, p-GST-Ana2^1−140aa^). The extra band disappeared upon *λ*PPase-treatment, proving that it was indeed Plk4-mediated phosphorylation of the serine residue within the ANST motif (Figure 4a). 

In a similar experimental setup, we tested the ability of MBP-Plk4-TE to modify the STAN motif in GST-Ana2^281−420aa^, a prerequisite for Sas6 binding to the protein [32]. We showed that in all cases, immobilized GST-Ana2^281−420aa^ treated with the purified kinases was capable of Sas6-binding, and it was independent of PP5’s activity (Figure 4b). This means, that the PP5-dephosphorylated MBP-Plk4-TE is able to modify the STAN motif, and promote Sas6-binding to the immobilized Ana2, in vitro. 

In all of the above experiments, we used the constitutively active recombinant Plk4. Therefore, it was reasonable to investigate how PP5 influences the function of the non-constitutively active Plk4, if at all. To this end, we co-expressed GFP-Plk4-ND or the kinase dead GFP-Plk4-NDKD, with Myc-PP5^E97Q^ (active) or Myc-PP5^H326N^ (inactive), respectively, in D.Mel-2 cells. Then we tested the Sas6-binding of GST-Ana2^281−420aa^ pre-treated with the GFP-Trap-purified ND or NDKD kinases, respectively. This in vitro assay revealed that PP5 only slightly influences the phosphorylation-dependent interaction of Ana2 and Sas6, i.e., the activity of the transgenic Plk4 derivatives (Figure 4). All of these data indicate that PP5-mediated dephosphorylation of Plk4 does not significantly influence the general activity of the kinase. 

### 2.5. PP5 Affects the Interaction of Plk4 with Asl and Other Centrosomal Proteins

Next, we tested the binding of the self-phosphorylated and dephosphorylated Plk4 to its partner Asterless (Asl), the scaffold of Plk4 in *Drosophila* [36]. To this end, we immobilized self-phosphorylated MBP-Plk4-TE on amylose resin and treated with *λ*-phosphatase, GST-PP5 or GST-PP5^H326N^, respectively, in the absence of ATP (to avoid re-self-phosphorylation of the kinase). Equal amounts of MBP (negative control), dephosphorylated and phosphorylated MBP-Plk4-TE immobilized on amylose resin (Figure 5) were incubated with ^35^S-labelled Asl generated by IVTT reaction. We observed that the *λ*-phosphatase as well as PP5-mediated dephosphorylation of MBP-Plk4-TE affects the interaction between Plk4 and Asl, in vitro (Figure 5). Dephosphorylated Plk4 strongly interacts with Asl, while its self-phosphorylated form (non-treated or incubated with inactive PP5^H326N^) shows less binding to Asl (Figure 5). We also purified and immobilized MBP-Plk4-TE co-expressed with either His_6_-tagged PP5 (dephosphorylates Plk4) or PP5^H326N^ (Plk4 remains phosphorylated) in *E. coli* cells, and tested their binding preference to Asl. This confirmed the above results and showed that dephosphorylated Plk4 binds more strongly to Asl.

We also investigated the binding of Plk4 to *Drosophila* Spd2 (served as the negative control), which is the centriolar partner of Zyg-1/Plk4 in *C. elegans* [37], but not in fruit flies, as well as Slimb (served as the positive control), which is the Plk4 phospho-degron-binding SCF subunit [33]. In addition, we included Ana2, Ana1, Sas6, Cep135, CP110, Polo and Sas4, putative regulators/interactors/substrates of Plk4 [29]. We co-expressed MBP-Plk4-TE with either His_6_-PP5 or PP5^H326N^ in *E. coli* cells, immobilized equal amounts of MBP (negative control), dephosphorylated and phosphorylated MBP-Plk4-TE on amylose resin (Appendix A) and incubated with ^35^S-labelled putative partner proteins generated by IVTT reactions. We found that *Drosophila* Spd2 bound to neither form of Plk4, while Slimb showed preference to the phosphorylated Plk4 derivative (Appendix A). Interestingly, in some cases, the interaction was significantly altered, while in other cases, slightly or not at all, depending on the phosphorylation-status of Plk4. We showed that Sas4, Cep135 and CP110 bound preferentially to dephosphorylated Plk4 (similarly to Asl), while Polo, Slimb and Ana1 bound rather to the self-phosphorylated form (Appendix A, p-MBP-Plk4-TE). Surprisingly, Ana2 and Sas6 bound to neither form of Plk4 in this assay. 

These in vitro results led us to speculate that PP5 might regulate the binding of Plk4 to its scaffold protein, Asl, and might affect the interaction with other centrosomal partner proteins. 

### 2.6. PP5 Interacts and Co-Localizes with Plk4 at the Centrosomes

Together, the above observations suggested that PP5 interacts with and dephosphorylates Plk4, which might happen at the centrosomes. To investigate the centrosomal interplay between Plk4 and PP5, we generated a highly specific antibody against *Drosophila* PP5. The antibody worked for western blotting (Appendix A), but not for immunostaining. Therefore, we followed two strategies to test the centrosomal localization of PP5. First, we determined the protein levels of PP5 and Plk4 throughout ontogenesis, and found that PP5 is present at high levels in all embryonic stages, and at moderate levels in larvae, pupa and adults (Appendix A). Plk4 was mostly observed in the 0–2 h-old syncytial embryos, the developmental stage of rapid and synchronized nuclear divisions (Appendix A). Therefore, we decided to prepare centrosomes from 0–2 h-old embryos by linear sucrose gradient ultracentrifugation and found that Plk4 and PP5 co-migrated with Asl (Plk4 partner in fruit flies) and *γ*Tubulin (centrosome marker) in the centrosome-enriched fractions (Figure 6). In contrast, αTubulin (non-centrosomal protein) appeared only in the cytoplasmic fractions. This provided evidence that a portion of the endogenous PP5 localizes to the centrosomes in *Drosophila* embryos.

To further test whether PP5 and Plk4 interact at the centrosomes, we employed the bimolecular fluorescence complementation (BiFC) assay [38], a useful technique to validate the direct interaction between proteins, also in *Drosophila* [39]. In this assay, the non-fluorescent split fragments of the YFP protein (NYFP and CYFP) are fused to the putative binding partners. If the candidates interact directly, the fully functional YFP is assembled from its split fragments and is then capable of generating the fluorescent signal. To this end, we transiently co-transfected mCherry-Sas6-expressing (centrosome marker) D.Mel-2 cultured cells with NYFP-Plk4 and CYFP-Asl (positive control), as well as NYFP-Plk4 and CYFP-PP5, respectively. Expression of the three metallothionein promoter-driven transgenes was induced with CuSO_4_ for 24 h, followed by live cell imaging to detect the YFP and mCherry signals. We observed that similarly to the positive control setup (NYFP-Plk4 and CYFP-Asl), NYFP-Plk4 and CYFP-PP5 interact with each other and co-localize with mCherry-Sas6 at the centrosomes in the triple transfected cells (Figure 7a). In the negative control setup (NYFP-Plk4 and CYFP alone, or CYFP-PP5 and NYFP alone), we could not detect green centrosomal signals (Appendix A). To avoid misinterpretation of the results, due to an otherwise unlikely intermolecular interaction between mCherry and the Split-YFP domains, we repeated this experiment in wild type D.Mel-2 cells co-expressing NYFP-Plk4 and CYFP-Asl (positive control) or NYFP-Plk4 and CYFP-PP5, respectively. These cells were fixed and stained with anti-Sas6 or anti-Asl, respectively, to visualize centrosomes. Similarly, to the above results, we could detect the green signals generated by NYFP-Plk4 and CYFP-Asl or NYFP-Plk4 and CYFP-PP5, respectively, at the centrosomes (Figure 7b). In the negative control setup, we did not see specific green signals in the centrosomes visualized by anti-Asl staining in far red (Appendix A). Together, these data strongly suggest that PP5 interacts with Plk4 at the centrosomes. 

### 2.7. PP5 Is Necessary to Maintain Normal Fertility

We assumed that PP5 might regulate the centrosomal function of Plk4, which would manifest in either a centrosome loss or gain upon inactivation of the phosphatase. To test this, we depleted the endogenous PP5 by transfecting D.Mel-2 cultured cells with dsRNA highly specific to PP5. Although the knock-down of the endogenous PP5 was efficient (Appendix A), it had only a mild effect on the centrosome numbers and mitotic index. This is in good agreement with the results of a high throughput RNAi screen, where phosphatases, including PP5, were tested for their cell cycle function in *Drosophila* cultured cells [22]. We also generated a *pp5^Δ^* null mutant *Drosophila* line using the CRISPR/Cas9 technique (Appendix A). Surprisingly, we found that hemizygotic flies bearing the *pp5^Δ^* null allele over different deficiencies are viable. Although we could not detect any obvious mitotic phenotypes in *pp5^Δ^/Df(3R)5454* embryos (hereafter *pp5^Δ^/Df*), we observed a significant fertility loss in adults. We found a reduced number of progeny from the cross of *pp5^Δ^/Df* females and *pp5^Δ^/Df* males, compared to the same *pp5^Δ^/Df* females crossed with wild type males or to maternally and zygotically wild type animals (Figure 8). This presumes a non-vital, but certainly important, zygotic function of PP5 in oogenesis and/or spermatogenesis. A western blot analysis of PP5 and Plk4 distribution in syncytial embryos and dissected ovaries or testes have shown that while PP5 is present in all three tissues, Plk4 is primarily present in ovaries (and in early embryos due to maternal effects), but barely detectable in testes (Appendix A). Because the fertilized embryo inherits the parental centrosome from the sperm [40], we investigated the morphology of the testes of *pp5^Δ^/Df* mutants. We could not detect obvious morphological abnormalities during spermatogenesis, and found normal basal body formation in the testes of *pp5^Δ^/Df* mutants (Appendix A). The above results strongly argue that the fertility issue of *pp5^Δ^/Df* null animals is a consequence of shared maternal and zygotic functions of PP5. It is currently unclear whether this phenomenon can be linked to (PP5-mediated) Plk4 function, as well, or it is completely independent from it. 

## 3. Discussion

Interestingly, the first discovered phosphatase, the PR enzyme, was published in 1945 by the Cori couple, 10 years before its opposing enzyme, the first kinase, phosphorylase b kinase, was discovered. Nevertheless, phosphatases have been much less studied over the decades for historical and technical reasons [41]. They had long been considered housekeeping enzymes without any explicit regulatory function. This eventually turned out to be incorrect as they are highly specific to different substrates, including kinases [31]. Since many kinases are encoded by oncogenes [30], it is not surprising that the counteracting enzymes are becoming recognized as potential targets for anticancer therapy [6]. 

The evolutionarily conserved PP5 phosphatase governs several vital cellular functions, including gene expression, DNA damage response, circadian clock, receptor signaling and apoptosis (reviewed in [5,6]). Moreover, its role in cell division has also been reported [20,22]. Therefore, we aimed to identify novel mitotic substrates of this enzyme and found that PP5 interacts with the Polo-like kinase 4, Plk4. Plk4 is an oncogenic kinase [30] and the master regulator of centriole duplication [24,25,26,27]. The maturation, activity, stability and localization of the Plk4 protein is regulated primarily by phosphorylation. Following synthesis, the enzyme has a low basic activity, which increases upon homo-dimerization of Plk4 and trans-auto-phosphorylation of the kinase domain [42,43]. Self-phosphorylation of the DRE element leads to Slimb/SCF-mediated proteasomal degradation of the kinase [33,44], which is reversed in mitosis by the phosphatase PP2A-Twins to stabilize Plk4 and promote centrosome duplication in the next S phase [31]. Interestingly, while other kinases also contribute to Plk4 regulation [30], the roles of other phosphatases have not yet been found.

We discovered that PP5 specifically interacts with Plk4 in fruit flies and human cells. We presented that the phosphatase domain of PP5 physically binds to the kinase domain and PB1 domains of Plk4 in *Drosophila*. We proved that Plk4 is dephosphorylated by PP5 in vitro and in vivo, and provided evidence that PP5 interacts and co-localizes with Plk4 at the centrosomes in *Drosophila*. We observed that the activity of Plk4 was not altered upon PP5 dephosphorylation, and the depletion of PP5 caused only a mild reduction in the centrosome number in cultured cells. However, we also found that the PP5-regulated dephosphorylation of Plk4 affects the interaction of Plk4 with various centrosomal proteins, including its scaffold protein, Asl, which preferentially binds to the dephosphorylated form of the kinase. Similarly, Plk4 binding to Sas4, the centriolar partner of Asl [36], requires dephosphorylated Plk4. In contrast, Ana1, the key protein in centriole-to-centrosome conversion [45] interacts exclusively with fully phosphorylated Plk4. Although we do not understand the mechanistic details of this regulation, it may be important for the centrosomal recruitment or maintenance of Plk4.

Considering the fact that PP5 is an evolutionarily conserved enzyme that regulates key determinants, such as tumor suppressor p53 [17], checkpoint kinases ATM, ATR [14,15,16] and Chk1 [46] or subunits of APC/C [20], we were surprised to see that its deletion from the *Drosophila* genome did not cause severe mitotic abnormalities or lethality. Indeed, the knock-out of PP5 in mice produced viable animals, too, with increased sensitivity to UV light [46]. This suggests that PP5 is not essential, but rather acts as a pleiotropic modulator of cellular processes in animals. PP5 is unique among the PPP family of Ser/Thr phosphatases, because it acts as a monomer and its low basal activity is self-regulated by its own TPR motifs [7], which are released by the molecular chaperone Hsp90 [12,47]. It has been shown that PP5 is a co-chaperone of Hsp90 and, together with Cdc37, forms a heteromeric complex [10] involved in the regulation of oncogenic kinases [11]. The González laboratory has proved that Hsp90 is a core centrosomal protein in *Drosophila* [48]. The above findings led us to speculate that the Hsp90-Cdc37-PP5 complex might be involved in the phospho-regulation of Plk4, which is not a vital modification. This may happen under special conditions or in defined cell cycle stages to ensure proper homeostasis of Plk4 at the centrosomes. Deciphering and answering this question promises to be an exciting future challenge.

## 4. Materials and Methods

### 4.1. DNA Constructs

cDNAs of the fruit fly PP5 (Clone ID: GH12714; Flybase ID: CG8402; *PpD3* in *Drosophila melanogaster*), CP110 (Clone ID: GH03511; Flybase ID: CG14617), Slimb (Clone ID: LD08669; Flybase ID: CG3412), Spd2 (Clone ID: LD24702; Flybase ID: CG17286), Polo (Clone ID: LD11851; Flybase ID: CG12306), Sas6 (Clone ID: FI21744, Flybase ID: CG15524), Asterless (hereafter Asl, Clone ID: GH02902, Flybase ID: CG2919) and Ana2 (Clone ID: LD22033, Flybase ID: CG8262) were obtained from the *Drosophila* Genomics Resource Centre (DGRC, Bloomington, IN, USA). cDNA of human PP5 (hereafter HsPP5, Clone ID: 3459309) and Plk4 (hereafter HsPlk4, Clone ID: 5273226) were obtained from Horizon Discovery Ltd (Cambridge, UK). Gateway entry clones of Ana1, Sas6, Ana2^1−140aa^ and Ana2^281−420aa^, the constitutively active form of the *Drosophila* Plk4 (hereafter Plk4-TE; Flybase ID: CG7186, contains the T172E mutation in the T-loop of the kinase) and its kinase dead version (hereafter Plk4-TEKM, kinase dead due to the K43M mutation), the non-degradable Plk4 (hereafter Plk4-ND, non-degradable due to the S293A and T297A mutations in the DRE region) and its kinase dead version (hereafter Plk4-NDKD, non-degradable and kinase dead due to the mutations S293A, T297A and K43M), and MBP-tagged Plk4-TE and Plk4-TEKM were obtained from the Glover lab (Cambridge, UK) [32,35]. Gateway entry clones of fruit fly Polo and PP5, as well as human PP5 were generated, according to the manufacturer’s instructions (cat # 11789100, Thermo Fisher Scientific, Waltham, MA, USA). The following destination vectors were used for the LR reaction to generate expression clones: for N-terminal GFP-tagging in *Drosophila* and human cells (pAGW (#1071 DGRC, Bloomington, IN, USA) and pcDNA-Dest53 (cat # 12288015, Thermo Fisher Scientific, Waltham, MA, USA), respectively); N-terminal Flag-tagging in *Drosophila* cells (pAFW (#1111, DGRC, Bloomington, IN, USA); N-terminal GST-tagging in *E. coli* (pDest15, cat # 11802014, Thermo Fisher Scientific, Waltham, MA, USA); and N-terminal His_6_-tagging in *E.coli* or IVTT (pDest17, cat # 11803012, Thermo Fisher Scientific, Waltham, MA, USA). 

Site-directed mutagenesis with the QuikChange II XL Mutagenesis Kit (cat # 200522, Agilent Technologies, Santa Clara, CA, USA) was performed using entry clones, according to the manufacturer’s instruction to generate truncated fruit fly PP5 and Plk4 proteins (PP5^203−520aa^, PP5^1−202aa^, Plk4-ΔKD, Plk4-ΔCC, Plk4-ΔDRE, Plk4-ΔPB1, Plk4-ΔPB2, Plk4-ΔPB1-2 and Plk4-ΔPB3); Plk4 fragments (KD, CC + DRE, L1 + PB1, PB1, PB1-2, PB2 and L2 + PB3) and amino acid substitutions in PP5 (fruit fly PP5^E97Q^ (hyperactive) and PP5^H326N^ (inactive), according to [7,49]) (see also Appendix A). PP5 truncated forms and point mutants were cloned to pDest15 and used in activity assay or GST-IVTT experiments as bait, while Plk4 truncated forms and fragments were cloned into pDest17 and used for ^35^S-labelling in IVTT reaction. cDNA of human Plk4 (hereafter HsPlk4, Clone ID: 5273226) was subjected to site-directed mutagenesis, as described above, to generate its non-degradable form (by introducing S285A and T289A mutations based on [33]), which was cloned to pFlag-CMV-4 plasmid (cat # E7158, Merck-Millipore, Burlington, MA, USA) to express Flag-HsPlk4-ND in human cells.

For co-expression in D.Mel-2 cells the Myc-tagged fruit fly PP5^E97Q^ or PP5^H326N^ and the GFP-tagged Plk4-ND or NDKD, respectively, were cloned to the Ac5-STABLE2-neo vector (Sutherland Lab, Bilbao, Spain [50]) by classical and InFusion cloning methods (cat # 102518, Takara Bio, Kusatsu, Siga, Japan). For co-overexpression in *E. coli* cells, the MBP-tagged Plk4-TE or TEKM (in pKM596 plasmid [32]) and His_6_-tagged PP5/PP5^H326N^, respectively, were used. To avoid replication origin incompatibility, we cloned the His_6_-tagged PP5/PP5^H326N^ to pACYCDuet-1 (cat # 71147-3, Merck-Millipore, Burlington, MA, USA). For the in vivo co-localization assay, we generated two new *Drosophila* Gateway vectors suitable for N-terminal tagging of any gene of interest with NYFP or CYFP, respectively, by modifying the original Split-YFP plasmids published earlier [39]: pMT(Hygro)-NYFP-Myc-GW and pMT(Hygro)-CYFP-HA-GW (Appendix A). CDS of Sas6 was cloned into the newly generated pMT(Hygro)-mCherry-GW (Appendix A) Gateway vector to establish stably transfected D.Mel-2 cell lines expressing mCherry-Sas6 in an inducible manner. The three new pMT vectors (sequences and maps are provided in Appendix A) are regulated by the metallothionein promoter (MT) suitable for copper-induced expression of the fusion proteins and hygromycin-selection in *Drosophila* cultured cells. 

For IVTT expression, we used Ana1, Sas4 [32], Cep135 [45], Polo and Plk4 constructs cloned to pDest17. Asl, Ana2, CP110 were IVTT synthesized directly from T7-promoter-driven cDNA clones obtained from DGRC; Sas6, Spd2 and Slimb were cloned to pHY22 vector [34]. 

All DNA constructs were validated by DNA sequencing. Oligonucleotide primers are provided in Appendix A.

### 4.2. Recombinant Protein Expression and Purification

Recombinant GST, GST-Ana2^1−140^, GST-Ana2^281−420^ and GST-tagged PP5 and its derivatives were expressed in SixPack *E. coli* [51] and purified following standard procedures. Briefly, bacteria were grown in Terrific broth auto-induction medium (cat # AIMTB0210, Formedium, Hunstanton, UK) for 48 h at 18 °C. Cells were lysed by sonication in 40 mL phosphate buffered saline (PBS) containing 0.2 mg/mL lysozyme and 1 mM PMSF (phenylmethylsulfonyl fluoride, cat # P7626, Sigma-Aldrich, St. Louis, MO, USA). Cleared supernatants were mixed with pre-equilibrated glutathione sepharose 4B resin (cat # 17-0756-01, Cytiva, Washington, WA, USA) and incubated at 4 °C for 1.5 h. Immobilized proteins were either kept on beads and stored at −20 °C in PBS supplemented with 50% glycerol or eluted with 10 mM reduced glutathione dissolved in phosphatase buffer (50 mM Na-HEPES pH 7.5, 100 mM NaCl, 2 mM DTT, 0.05% Triton X-100, 2 mM MgCl_2_, 2 mM MnCl_2_). Amicon Ultra-0.5 Centrifugal Filter Units (cat # UFC503096, Merck-Millipore, Burlington, MA, USA) were used to concentrate the eluted proteins, which were then supplemented with 50% glycerol and stored at −80 °C. His_6_-tagged PP5 was expressed in SixPack and purified on Ni-sepharose 6 Fast Flow resin (cat # 17531802, Cytiva, Washington, WA, USA), according to the manufacturer. Eluted proteins were dialyzed over PBS for 18 h at 4 °C, concentrated and used for mice immunization. 

Bacteria expressing MBP-Plk4-TE or MBP-Plk4-TEKM, or co-expressing MBP-Plk4-TE or MBP-Plk4-TEKM with His_6_-PP5 or His_6_-PP5^H326N^, respectively, were grown in LB medium and induced with 1 mM isopropyl 1-thio-β-D-galactopyranoside (cat # R0393, Thermo Fisher Scientific, Waltham, MA, USA) for 2 h at 25 °C, and another 4 h at 30 °C. Cell lysates were prepared by sonication using Branson Sonifier (Branson Ultrasonics Corp., Danbury, CT, USA) in PBS supplemented with 100 mM NaCl, 1 mM PMSF, EDTA-free protease inhibitor cocktail (cat # 11873580001, Roche, Basel, Switzerland), and 0.05% Triton X-100. Cleared supernatants were mixed with pre-equilibrated amylose resin (cat # E8021S, New England Biolabs, Ipswich, MA, USA) and incubated at 4 °C for 1.5 h. Beads were washed with PBS supplemented with 100 mM NaCl and 0.05% Triton X-100 followed by washing with kinase buffer (20 mM Na-HEPES pH 7.5, 100 mM NaCl, 10 mM MgCl_2_, 10 mM MnCl_2_, 1 mM DTT). Immobilized MBP-tagged proteins were either kept on beads and stored at −20 °C in DTT-free kinase buffer supplemented with 50% glycerol, or eluted with kinase buffer containing 15 mM D-maltose. Eluted proteins were concentrated, then supplemented with 50% glycerol and stored at −80 °C. For the in vivo phosphatase assay bacteria co-expressing MBP-Plk4-TE or MBP-Plk4-TEKM with His_6_-PP5 or His_6_-PP5^H326N^, respectively, were boiled in Laemmli sample buffer (5 min 100 °C). 

### 4.3. IVTT and In Vitro Binding Assay

For testing the in vitro physical protein-protein interactions, we produced the ^35^S-methionine-labeled candidate binding partners in a coupled in vitro transcriptional/translational system (IVTT). The detailed protocol of this method is provided in [34]. Briefly, bait proteins were immobilized either on glutathione sepharose 4B beads (GST, GST-PP5, GST-PP5^H326N^, GST-PP5^1−202aa^, GST-PP5^203−520aa^, GST-PP5^E97Q^, GST-Ana2^281−420aa^ and GST-Ana2^1−140aa^) or amylose resin (MBP or MBP-Plk4-TE), and incubated with ^35^S-labelled putative partners. Protein samples were subjected to SDS-PAGE followed by Coomassie brilliant blue-staining. Gels were scanned, dried and the specific protein-protein interactions were detected by autoradiography. 

### 4.4. Drosophila and Human Cultured Cell Lines—Transfection and Maintenance

D.Mel-2 cells (Schneider’s *Drosophila* Line 2 [D. Mel. (2), SL2] (ATCC CRL-1963, Manassas, VA, USA) were cultured in Insectagro DS2 Serum-Free medium (cat # 13-402-CV Corning, Corning, NY, USA) supplemented with 2 mM stable L-glutamine (cat # XC-T1755, Biosera, Nuaille, France) and 1× PenStrep (cat # XC-A4122, Biosera, Nuaille, France) at 25 °C. Cells were transfected using Expifectamine Sf reagent (cat # A38915, Thermo Fisher Scientific, Waltham, MA, USA), according to the manufacturer. Transiently transfected cells were harvested 2 days-post transfections (if metallothionein promoter was used, cells were treated with 0.5 mM CuSO_4_ for 24 h before harvesting). For the generation of stable transfected D.Mel-2 lines, cells were subjected 3 days-post transfection to antibiotic selection (following standard procedures) for 4 weeks using G418 at 1 mg/mL concentration (cat # A1720-5G, Merck-Millipore, Burlington, MA, USA) or hygromycin B at 300 µg/mL concentration (cat # 25965.03, Serva, Heidelberg, Germany), respectively.

Human HEK293 cells (ATCC CRL-1573, Manassas, VA, USA) were cultured in DMEM containing GlutaMAX™ Supplement (cat # 61965026, Thermo Fisher Scientific, Waltham, MA, USA) and supplemented with 10% FBS (cat # ECS0180L, Euroclone, Pero, Italy), 1x PenStrep and NEAA mixture (cat # BE13-114E, Lonza, Basel, Switzerland) at 37 °C under 5% CO_2_. Cells were transiently co-transfected using PEI (cat # 408727, Merck-Millipore, Burlington, MA, USA), according to standard procedures, harvested 2 days post-transfection, then processed for co-immunoprecipitation.

### 4.5. Gene Silencing in Drosophila Cultured Cells 

RNAi approach was used to test the specificity of the anti-PP5 polyclonal antibody. dsRNAs targeting the bacterial kanamycin gene (negative control) or the *PpD3* gene encoding PP5 (either the coding sequence (CDS) or 3′-UTR) were designed. dsRNAs were synthetized using the MEGAscript T7 transcription kit (cat # AM1334, Thermo Fisher Scientific, Waltham, MA, USA), according to the manufacturer. dsRNA-treatment was performed, as described earlier [32]. Briefly, D.Mel-2 cells were cultured in 6-well plates and transfected with 10 μg dsRNA using TransFast reagent (cat # E2431, Promega, Madison, WI, USA) for 3 days at 25 °C. Oligonucleotide primers used in this study are shown in Appendix A.

### 4.6. Protein Sample Preparation from the Tissues

D.Mel-2 cells transiently co-transfected with GFP or GFP-PP5 and 3×Flag-Plk4-ND were lysed by 10 × passing the cell suspension through a G25 needle in EB buffer (20 mM Tris-HCl pH 7.6, 150 mM NaCl, 0.5 mM EGTA, 2 mM MgCl_2_, 0.1% NP40, 5% glycerol, 1 mM DTT, 1 mM PMSF, EDTA-free protease inhibitor cocktail, 0.1 µL/mL benzonase nuclease (cat # 70746-3, Merck-Millipore, Burlington, MA, USA). Lysates were centrifuged (17,000× *g*, 20 min, 4 °C) and cleared supernatants were used for immunoblotting and co-immunoprecipitations. 

Stably transfected D.Mel-2 cells expressing GFP-Plk4-ND or GFP-Plk4-NDKD and Myc-PP5^E97Q^ or Myc-PP5^H326N^, respectively, were directly lysed in Laemmli sample buffer containing 0.1 µL/mL benzonase nuclease (treated for 5 min at room temperature followed by boiling for 5 min) and used for immunoblotting; or, they were lysed in EB supplemented with PhosSTOP phosphatase inhibitor cocktail (cat # 4906837001, Merck-Millipore, Burlington, MA, USA) by the needle/syringe method, and used for GFP-Trap (cat # gtma, ChromoTek GmbH, Planegg, Germany) purification. The captured kinases were used for the in vitro kinase assay to phosphorylate GST-Ana2^281−420aa^, in vitro.

HEK293 cells were transiently co-transfected with GFP or GFP-HsPP5 and Flag-HsPlk4-ND, respectively, lysed in EB by passing the cell suspension through a G25 needle. Lysates were centrifuged (17,000× *g*, 20 min, 4 °C) and cleared supernatants were used for immunoblotting and co-immunoprecipitations. 

Total protein extracts were prepared from *Drosophila* wild type embryos (0–24 h-old), L1 to L3 larvae, white and brown pupae, pharate adult animals, and adults, respectively, by grinding the samples in RIPA buffer (10 µL buffer for 1 mg sample); 50 mM Tris-HCl pH 8.0, 150 mM NaCl, 1% NP 40, 0.5% Na-Deoxycholate, 0.1% SDS, 1 × EDTA-free protease inhibitor cocktail supplemented with 0.1 µL/mL benzonase nuclease. Samples were incubated on ice for 20 min, then supplemented with Laemmli sample buffer, boiled for 5 min and subjected to SDS-PAGE and immunoblotting. *Drosophila* syncytial embryos (0–2 h-old), and dissected ovaries and testes were collected in Laemmli sample buffer, boiled for 5 min, homogenized by glass pestles and boiled for another 5 min. Equal amounts of samples were subjected to SDS-PAGE and immunoblotting. 

For antibody validation, we homogenized the *pp5* null mutant adult flies in Laemmli sample buffer with a glass pestle and boiled the samples for 5 min before they were subjected to immunoblotting. Cultured cells depleted for the endogenous PP5 by RNAi were harvested 3 days post-transfection with dsRNAs, washed with PBS then boiled in Laemmli sample buffer for 5 min. Equal amounts of protein samples were subjected to immunoblotting.

### 4.7. Co-Immunoprecipitation 

Co-immunoprecipitation from cultured *Drosophila* cells was carried out according to [52]. Briefly, D.Mel-2 cells expressing GFP or GFP-PP5 and 3×Flag-Plk4-ND were harvested 2 days post-transfection, washed in sterile PBS and lysed in EB buffer. Clarified supernatants were used for purification on GFP-Trap magnetic agarose beads (cat # gtma-20, ChromoTek GmbH, Planegg, Germany) at 4 °C for 90 min. Bound proteins were eluted by boiling the beads in Laemmli sample buffer, then fractionated on SDS-PAGE and blotted onto a PVDF membrane. Immunoprecipitation was tested by immunoblotting with anti-GFP and anti-FlagM_2_ antibodies, respectively.

Stably transfected D.Mel-2 cells expressing GFP-Plk4-ND or GFP-Plk4-NDKD and Myc-PP5^E97Q^ or Myc-PP5^H326N^, respectively, were lysed in EB supplemented with PhosSTOP phosphatase inhibitor cocktail. GFP-tagged proteins were captured from clarified supernatants using GFP-Trap magnetic agarose beads and used for an in vitro kinase assay to phosphorylate GST-Ana2^281−420aa^.

For testing the in vivo interaction between human Plk4 and PP5, HEK293 cells were transiently transfected with GFP or GFP-HsPP5 and Flag-HsPlk4-ND, respectively. Cells were harvested 2 days post-transfection and lysed in EB, as above. GFP or GFP-HsPP5 were captured using GFP-Trap magnetic agarose beads at 4 °C for 90 min and eluted by boiling in Laemmli sample buffer, followed by SDS-PAGE fractionation and immunoblotting with anti-FlagM_2_ and anti-GFP antibodies.

### 4.8. In Vitro Kinase and Phosphatase Assay

In vitro kinase assay was carried out according to [32]. Briefly, 6 μg recombinant purified MBP-Plk4-TE (expressed either alone or in combination with His_6_-PP5 or PP5^H326N^ in *E. coli*) was used in 50 μL kinase reaction containing 250 μM ATP and 5 μg GST-Ana2^1−140 aa^ immobilized on glutathione sepharose 4B beads. The reaction was performed at 30 °C for 1 h, then the beads were washed once with λ-phosphatase buffer and divided into two equal parts: one part was untreated, while the other part was treated with 0.3 μL λ-phosphatase (cat # P0753S, New England Biolabs, Ipswich, MA, USA) for 30 min at 30 °C. The reaction was terminated by boiling in Laemmli sample buffer. Proteins were fractionated on SDS-PAGE and subjected to Coomassie brilliant blue-staining. 

The STAN-motif-phosphorylating activity of purified MBP-Plk4-TE (expressed either alone or in combination with His_6_-PP5 or PP5^H326N^ in *E. coli*) was tested in vitro in kinase reaction containing 250 μM ATP and 3 μg GST-Ana2^281−420aa^ immobilized on glutathione sepharose 4B beads. The reaction was performed at 30 °C for 1 h, followed by washing the beads with 50 mM Na-HEPES pH 7.5, 150 mm NaCl, 2 mm MgCl_2_, 1 mm EGTA, 1 mm DTT, 0.1% Triton X-100 and PhosSTOP phosphatase inhibitor cocktail. Then, the beads were incubated with ^35^S-labeled Sas6 generated in IVTT reaction for 1 h at 4 °C, and finally boiled in Laemmli sample buffer. Protein samples were run on SDS-PAGE, Coomassie brilliant blue-stained gels were dried and subjected to autoradiography.

GFP-Plk4-ND or NDKD co-expressed with Myc-PP5^E97Q^ or PP5^H326N^, respectively, in D.Mel-2 cells were captured on GFP-Trap magnetic agarose beads as above, and directly used in 50 μL kinase reaction containing 250 μM ATP and 3 μg purified GST-Ana2^281−420aa^ for 1 h at 30 °C. Then, pre-phosphorylated GST-Ana2^281-420aa^ was immobilized on glutathione sepharose 4B beads in the presence of PhosSTOP phosphatase inhibitor cocktail and incubated with ^35^S-Sas6 for 1 h at 4 °C. Protein samples were subjected to SDS-PAGE followed by Coomassie brilliant blue-staining. Dried gel was subjected to autoradiography. 

In vitro phosphatase assay was carried out in 20 μL phosphatase buffer containing 2 μg purified MBP-Plk4-TE and 5 μg purified GST-PP5 or GST-PP5^H326N^ at 30 °C for 1 h; 0.3 μL λ-phosphatase served as positive control and non-treated MBP-Plk4-TE served as negative control. Protein samples were boiled in Laemmli sample buffer, separated by SDS-PAGE and subjected to Coomassie brilliant blue-staining. To test the kinetics of in vitro dephosphorylation, we used 32 μg purified MBP-Plk4-TE in 160 μL phosphatase buffer, and removed 20 µL (for non-treated control). Then we added 5 μg GST-PP5 to the kinase and the reaction was performed at 30 °C. Then, 20 μL aliquots were taken at the indicated time points (5–60 min), supplemented with Laemmli sample buffer, and boiled for 5 min. Then, 4 μg purified kinase was treated with 0.3 μL λ-phosphatase (control treatment) for 60 min at 30 °C. Protein samples were separated on SDS-PAGE and subjected to Coomassie brilliant blue-staining. 

For testing the activity of purified GST-PP5 and its derivatives, we used p-nitrophenyl phosphate (pNPP) as a general chromogenic phosphatase substrate. The reaction was performed in 1 mL phosphatase buffer containing 2 μg purified proteins and 2,5 μL pNPP (500 mM, cat # P0757S, New England Biolabs, Ipswich, MA, USA) at 30 °C for 10 min, then the absorbance of the liberated p-nitrophenol (its amount is proportional with the phosphatase activity) was measured at 405 nm using NanoDrop OneC spectrophotometer (cat # ND-ONE-W, Thermo Fisher Scientific, Waltham, MA, USA). 

### 4.9. Centrosome Purification with Sucrose Gradient Ultracentrifugation

0.2 g of wild type (*w^1118^*) 0–2 h-old syncytial *Drosophila* embryos were collected, dechorionated and lysed in 1 mL BRB80 buffer (80 mM PIPES, 1 mM EGTA, 1 mM MgCl_2_ pH 6.8) using Dounce homogenizer (cat # 357542, VWR, Radnor, PA, USA) according to [53]. The lysate was clarified by mild centrifugation (1500× *g*, 10 min at 4 °C) and filtered through Miracloth (cat # 475855, Merck Millipore, Burlington, MA, USA); 0.4 mL filtrate was loaded onto a 5 mL sucrose gradient (25–70 *w*/*v* % linear sucrose gradient in BRB80 buffer supplemented with 100 mM KCl). Centrosomes were separated by ultracentrifugation (131,000× *g* in Beckman-Coulter (Brea, CA, USA) MLS-50 rotor for 90 min at 4 °C), fractions (0.2 mL each) were collected from the bottom of the tube and subjected to SDS-PAGE analysis. Immunoblotting was performed by using anti-γTubulin and anti-Asl as centrosome fraction markers (centrosomes enriched around 55–60% of sucrose) and anti-αTubulin as cytoplasmic fractions (around 25–33% of sucrose) marker. The co-migration and centrosomal co-localization of Plk4 and PP5 were also tested.

### 4.10. Antibodies

The anti-PP5 polyclonal antibody was generated in-house by immunizing BALB/c mice with purified His_6_-PP5 following standard procedures. The specificity of the antibody was confirmed by immunoblotting (Appendix A). The following antibodies were used in immunoblot (IB) or immunofluorescence (IF) experiments: mouse anti-PP5 (IB: 1:10,000), rabbit anti-Asl (IB: 1:2000, IF: 1:300 [36]), rat anti-Sas6 (IF: 1:500 [32]), sheep anti-Plk4 (IB: 1: 2000 [32], anti-γTubulin (IB: 1: 10,000, cat # T6557, Merck Millipore, Burlington, MA, USA), anti-αTubulin (IB: 1: 10,000, cat # T6199, Merck Millipore, Burlington, MA, USA), anti-GFP (IB: 1: 1000, cat # 11814460001, Roche, Basel, Switzerland), anti-Actin (IB: 1:2000, cat # A4700, Merck Millipore, Burlington, MA, USA), anti-FlagM_2_ (IB: 1:10,000, cat # F1804, Merck Millipore, Burlington, MA, USA), goat anti-mouse IgG conjugated to horseradish peroxidase (1:10,000, cat # P044701-2 Dako, Glostrup, Denmark), goat anti-rabbit IgG conjugated to horseradish peroxidase (1:10,000, cat # P044801-2, Dako, Glostrup, Denmark), goat anti-sheep IgG conjugated to horseradish peroxidase (1: 10,000, cat # 31480, Thermo Fisher Scientific, Waltham, MA, USA), donkey anti-Rabbit IgG Alexa Fluor 647 (IF: 1:300, cat # A31573, Thermo Fisher Scientific, Waltham, MA, USA), and donkey anti-Rat IgG DyLight 650 (IF: 1:500, cat # SA5-10029, Thermo Fisher Scientific, Waltham, MA, USA).

### 4.11. Microscopy

NYFP-Plk4 and CYFP (negative control), CYFP-PP5 and NYFP (negative control), NYFP-Plk4 and CYFP-Asl (positive control) or NYFP-Plk4 and CYFP-PP5, respectively, were transiently co-expressed in D.Mel-2 cells expressing the mCherry-Sas6 centrosomal marker protein under the control of MT promoter in glass bottom dishes (cat # P35G-0-14-C, Mattek Corporation, Ashland, MA, USA). Transgene expression was induced by adding 0.5 mM CuSO_4_ to the medium for 24 h. Then, 0.33 µg/mL Hoechst 33342 (cat # 62249, Thermo Fisher Scientific, Waltham, MA, USA) was added to the medium for 30 min to visualize DNA. Then the media was removed and cells were kept in mounting medium (cat # 50001, Ibidi, Gräfelfing, Germany) during microscopy. Images from live cells were taken with a Zeiss LSM 800 confocal microscope (Carl Zeiss, Jena, Germany) using an 63× OIL objective. YFP fluorescence signal of the samples was detected on the GFP channel with the same settings. mCherry signal was detected on the mCherry channel with the same settings.

D.Mel-2 cells co-expressing NYFP-Plk4 and CYFP (negative control), CYFP-PP5 and NYFP (negative control), NYFP-Plk4 and CYFP-Asl (positive control) or NYFP-Plk4 and CYFP-PP5, respectively, were collected, fixed and stained with DAPI, anti-Sas6 or anti-Asl antibodies according to [32]. Experiments were repeated four times. Centrosomal localization of the YFP signal was counted in *n* = 100 transfected cells. Images were taken using Olympus Fluoview Fv10i Confocal microscope (Olympus Corp., Tokyo, Japan).

Preparation of testis, fixation and staining with DAPI and Phalloidin were performed as described earlier [54]. Samples were mounted using SlowFade Gold antifade reagent (cat # S36967, Thermo Fisher Scientific, Waltham, MA, USA) and images were taken using Olympus Fluoview Fv10i Confocal microscope. 

### 4.12. Fly Stocks, Mutants and Fertility Test

*Drosophila melanogaster* flies were crossed and maintained on standard cornmeal agar medium at 25 °C. CRISPR/Cas9-mediated gene deletion was performed to generate the *pp5^Δ7/1/2^* null mutant (hereafter *pp5^Δ^*) following standard procedures [55]. Two guide RNAs (Appendix A) were designed to target the 5′-end of the second exon and 3′-UTR region of the PP5-encoding *PpD3* gene (FlyBase ID: CG8402). The deletion removed a 1936 bp-long sequence from the gene, which was validated by DNA sequencing (breakpoints: 3R:9748626 and 3R:9750562). The fertility of the animals was tested by crossing the appropriate genotype of individual females (*n* = 8) with three wild type (wt) or *pp5/Df(3R)ED5454* males. Experiments were repeated three times and the significance was determined by Welch’s test. Testes were analyzed in flies expressing the basal body marker GFP-PACT in wild type or *pp5/Df(3R)ED5454* hemizygotic genetic background. Stock used in this study:

*w^1118^* (BDSC ID: 5905)—isogenic wild type (wt) control stock.

*Df(3R)ED5454* (BDSC ID: 86547)—isogenic deficiency stock with chromosomal deletion over *PpD3*.

*Df(3R)ED5428* (BDSC ID: 9227)—isogenic deficiency stock with chromosomal deletion over *pPD3*.

*w; +; pp5^Δ7/1/2^/TM6bTbHu*—PP5 null mutant stock.

*w; Plp^PACT^.^Ubi-p63E^.^GFP^*—expressing GFP-PACT, a basal body marker in testis [56].

## Figures and Tables

**Figure 1 ijms-24-02033-f001:**
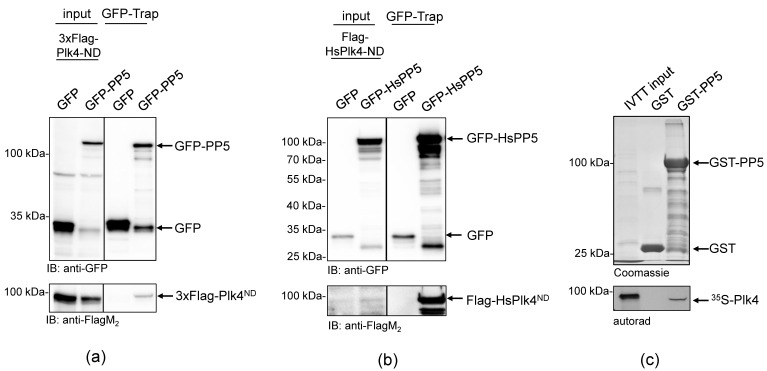
PP5 binds directly to Plk4 in *Drosophila* and human cells. (**a**) GFP-Trap affinity purification of GFP (negative control) or GFP-PP5 transiently co-expressed with Flag-Plk4-ND in D.Mel-2 cultured cells. Lysate inputs and purified proteins were subjected to SDS-PAGE and a western blot analysis, which shows that GFP-PP5 interacts with Flag-Plk4-ND. (**b**) GFP-Trap affinity purification of GFP (negative control) or GFP-HsPP5 (human PP5) transiently co-expressed with Flag-HsPlk4-ND in HEK293 cells. Western blot analysis shows that Flag-HsPlk4-ND specifically co-purifies with HsPP5 from human cells. (**c**) GST-IVTT in vitro binding assay: immobilized GST (negative control) or GST-PP5 recombinant proteins were incubated with ^35^S-Plk4 and the interaction was analyzed by SDS-PAGE followed by autoradiography (autorad). Plk4 directly and specifically binds to GST-PP5, in vitro.

**Figure 2 ijms-24-02033-f002:**
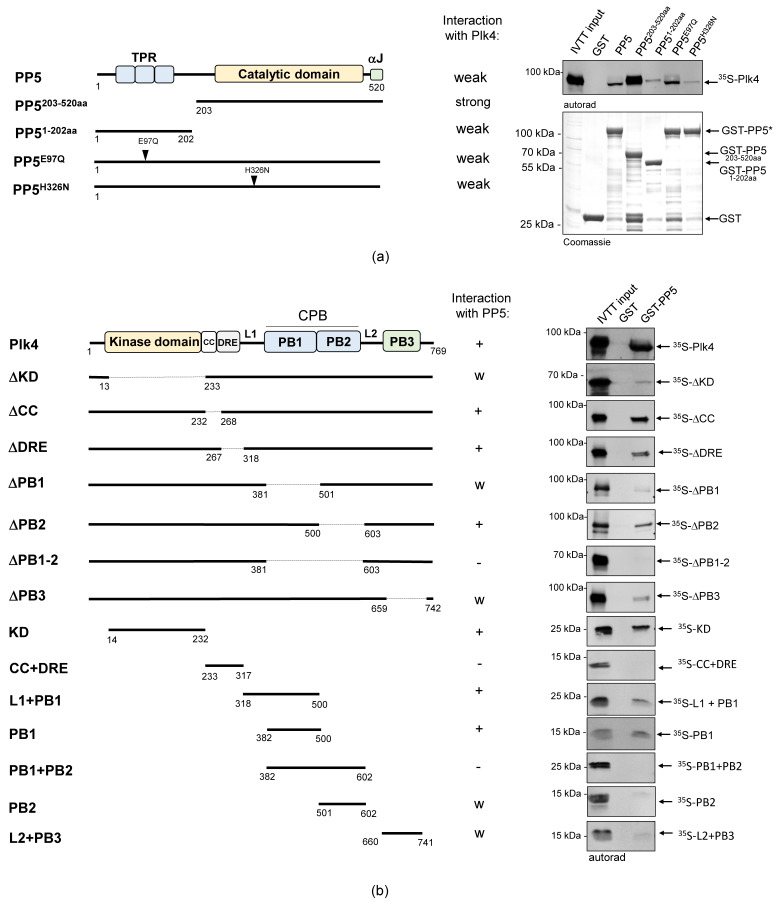
The catalytic domain of PP5 interacts with the kinase domain and PB1 domain of Plk4. (**a**) GST-IVTT in vitro binding assay: autoradiogram (right panel) shows that in the absence of the TPR-domain (catalytic domain only), GST-PP5^203−520aa^ binds more strongly to ^35^S−Plk4, in vitro. The wild type and hyperactive GST-PP5/PP5^E97Q^ bind moderately, while the inactive GST-PP5^H326N^ and the TPR domain-containing GST-PP5^1−202aa^ bind weakly to ^35^S-Plk4. The Coomassie brilliant blue-stained gel shows the loading of the bait proteins. The asterisk indicates the location of the wild type and point-mutant forms of GST-PP5. GST serves as the negative control. (**b**) Domain mapping of Plk4 was carried out with a GST-IVTT in vitro binding assay. Right panels (autoradiograms) show that GST-PP5 specifically binds to the kinase (KD) and PB1 domains of ^35^S-Plk4. GST serves as the negative control. “+” means strong binding; “−” means no binding; “w” means weak binding; “Δ” refers to the missing domain/motif of the full length protein (indicated by dashed lines in the diagram); numbers show the position of amino acid endpoints of the truncated proteins. Coomassie brilliant blue-stained gels (loading) are shown in Appendix A. Schematics on the left represent the domain/motif architecture and sizes of PP5 and Plk4 constructs used in this assay.

**Figure 3 ijms-24-02033-f003:**
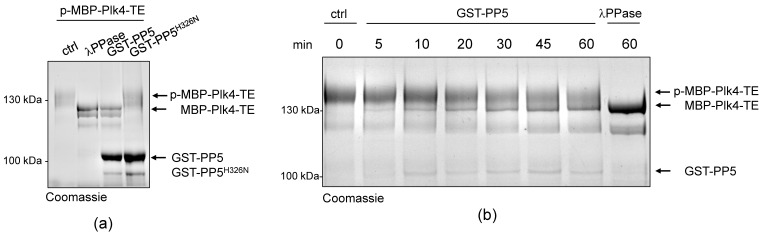
PP5 dephosphorylates Plk4, in vitro and in vivo. (**a**) In vitro dephosphorylation of p-MBP-Plk4-TE (“p-” refers to the self-phosphorylated, while “TE” refers to the T172E activating mutation in Plk4, respectively) was tested by gel-shift assay. The electrophoretic mobility of the phosphorylated forms is decreased, therefore they run higher, while the dephosphorylated species run lower in a long preparative SDS-PAGE. Purified recombinant p-MBP-Plk4-TE is efficiently dephosphorylated by the *λ*-phosphatase (*λ*PPase) and GST-PP5, in vitro. GST-PP5^H326N^ serves as the negative control; “ctrl” refers to non-treated purified kinase. (**b**) The GST-PP5-mediated dephosphorylation kinetics of p-MBP-Plk4-TE is presented by Coomassie brilliant blue-stained preparative SDS-PAGE. Incubation times (minutes) are indicated. *λ*PPase serves as the positive control; “ctrl” refers to non-treated sample. (**c**) In vivo dephosphorylation of Plk4 in bacteria was tested by gel-shift assay. MBP-Plk4-TE or its kinase dead (TEKM) version were co-expressed with His_6_-PP5 or its inactive form, His_6_-PP5^H326N^, respectively, in bacteria. Crude cell lysates were analyzed by SDS-PAGE, which shows that MBP-Plk4-TE dephosphorylation requires the active His_6_-PP5, in vivo. MBP-Plk4-TEKM is not capable of self-phosphorylation. His_6_-PP5^H326N^ is unable to dephosphorylate Plk4.

**Figure 4 ijms-24-02033-f004:**
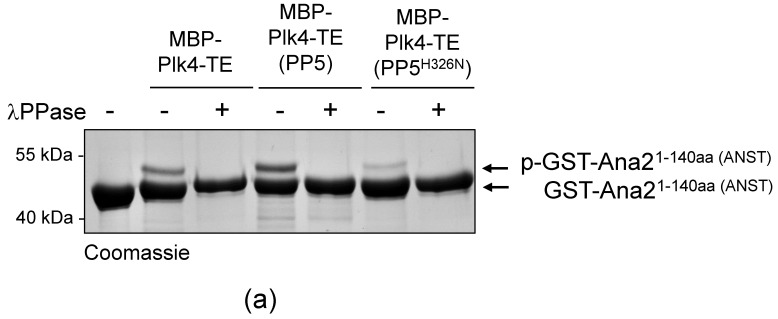
PP5 does not regulate the Plk4 kinase activity. (**a**) The in vitro phosphorylation of Ser38 within the ANST motif of GST-Ana2^1−140aa^ was analyzed by gel-shift assay. Constitutively active MBP-Plk4-TE expressed alone or co-expressed with either the wild type (PP5) or inactive (PP5^H326N^) His_6_-tagged PP5 in bacteria were purified and used to phosphorylate GST-Ana2^1−140aa^, in vitro. The gel-shift assay shows that all versions of MBP-Plk4-TE were able to modify the GST-Ana2^1−140aa^ protein, manifested as a slowly migrating phosphorylated band, which disappeared upon *λ*PPase-treatment. “+” means *λ*PPase-treatment, “−” means no treatment with *λ*PPase. (**b**) Sas6-binding to phosphorylated GST-Ana2^281−420aa^ was tested by GST-IVTT binding assays. Constitutively active MBP-Plk4-TE expressed alone or co-expressed with either the wild type (PP5) or inactive (PP5^H326N^) His_6_-tagged PP5 in bacteria were purified and used to phosphorylate the STAN motif in GST-Ana2^281−420aa^ immobilized on beads. The autoradiogram (autorad) shows that ^35^S-Sas6 interacts with p-GST-Ana2^281−420aa^, which means that in all cases Plk4 was able to phosphorylate the STAN motif. (**c**) GFP-Plk4-ND or GFP-Plk4-NDKD co-expressed with Myc-tagged active PP5^E97Q^ and inactive PP5^H326N^, respectively, were purified from D.Mel-2 cells and incubated with immobilized recombinant GST-Ana2^281−420aa^, to test whether the Ana2 derivative can bind to ^35^S-Sas6. Only the active GFP-Plk4-ND kinase was able to phosphorylate GST-Ana2^281−420aa^ and promote its binding to ^35^S-Sas6, which was independent of the activity of PP5. MBP-Plk4-TE was used as the positive control and GFP was used as the negative control. “p-” refers to the phosphorylated form of the protein.

**Figure 5 ijms-24-02033-f005:**
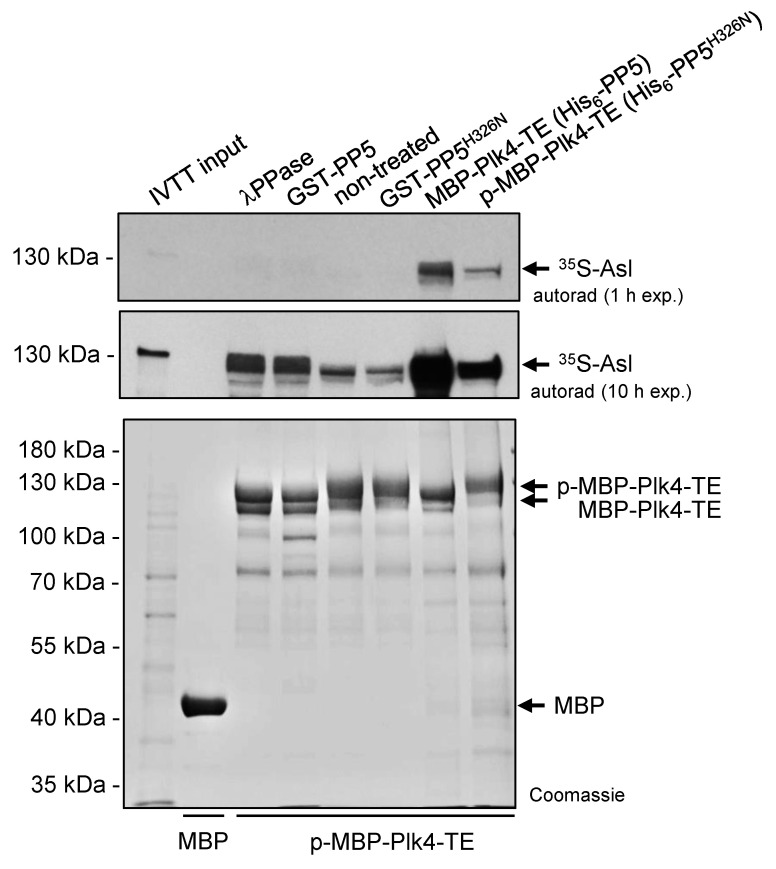
Asl preferentially binds to the PP5-dephosphorylated form of Plk4. Immobilized MBP-Plk4-TE was non-treated (lane 5) or treated with *λ*-phosphatase (*λ*PPase, lane 3), GST-PP5 (lane 4) or GST-PP5^H326N^ (lane 6), respectively, and tested for binding to ^35^S-Asl synthesized in IVTT reaction. MBP-Plk4-TE co-expressed with either His_6_-PP5 (wild type) or His_6_-PP5^H326N^ (inactive) in bacteria were immobilized to amylose resin and incubated with ^35^S-Asl (lane 7–8). The in vitro binding assay reveals that dephosphorylated MBP-Plk4-TE binds more Asl, while the self-phosphorylated kinase binds less Asl. MBP was used as the negative control (lane 2). “p-” refers to phosphorylated Plk4. Coomassie-stained gel shows the loading and shifting of MBP-Plk4-TE derivatives. Short (1 h) and long (10 h) exposures of the autoradiographs are presented.

**Figure 6 ijms-24-02033-f006:**
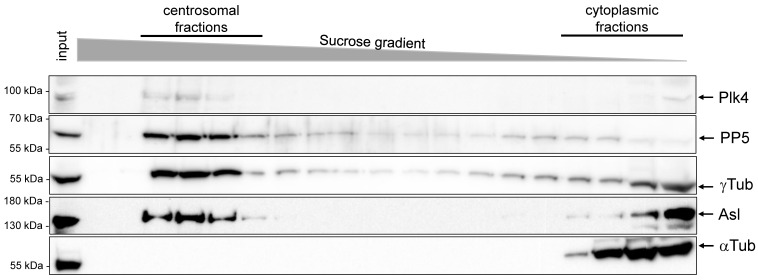
PP5 and Plk4 co-migrate with centrosomal proteins. A western blot analysis of the centrosomal and cytoplasmic fractions of proteins that were prepared from *Drosophila* syncytial embryos by sucrose gradient ultracentrifugation. Endogenous proteins were revealed by specific-antibodies (indicated on the right), which shows that PP5 co-fractionated with Plk4, as well as with the centrosomal markers Asl and *γ*Tubulin (*γ*Tub). αTubulin (αTub) represents the cytosolic fractions.

**Figure 7 ijms-24-02033-f007:**
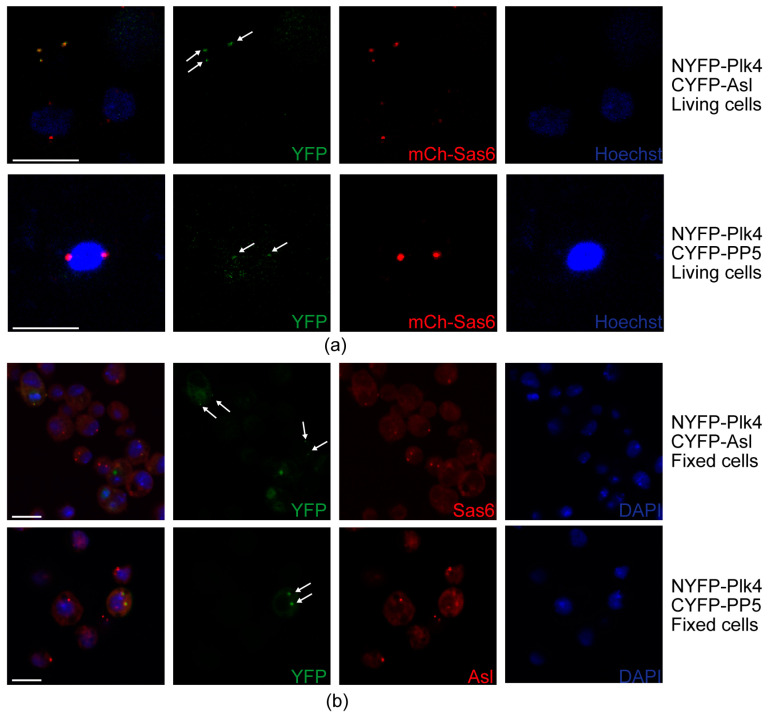
Interaction and co-localization of PP5 and Plk4 at the centrosomes. (**a**) Stills of live cell imaging of D.Mel-2 cells show that the co-expressed NYFP-Plk4 and CYFP-Asl (positive control), as well as NYFP-Plk4 and CYFP-PP5, co-localize (generate the green signal) to some of the mCherry-Sas6 (mCh-Sas6)-labelled centrosomes. Arrows indicate the co-localization. Co-expressed NYFP-Plk4 and CYFP, as well as CYFP-PP5 and NYFP, were used as negative controls (shown in Appendix A). Hoechst 33342 was used to visualize DNA. Scale bar: 10 µm (**b**) Representative microscopic images of fixed D.Mel-2 cells show that the co-expressed NYFP-Plk4 and CYFP-Asl (positive control), as well as NYFP-Plk4 and CYFP-PP5, interact and co-localize (generate the green signal) at the centrosomes (stained with anti-Sas6 or anti-Asl in far red, respectively). Co-localization was observed in 98% of the transfected cells (*n* = 100 transfected cells). Arrows indicate the co-localization. Co-expressed NYFP-Plk4 and CYFP, as well as CYFP-PP5 and NYFP, were used as negative controls (shown in Appendix A). DAPI was used to visualize DNA. Scale bar: 10 µm.

**Figure 8 ijms-24-02033-f008:**
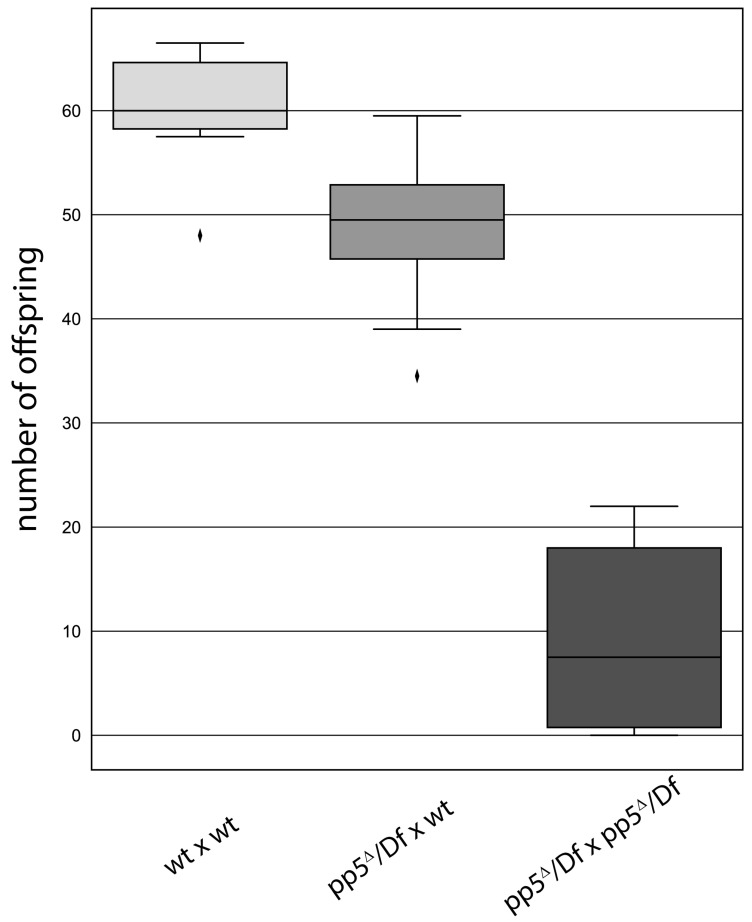
*pp5* deletion is viable, but leads to a reduced fertility in *Drosophila*. Wild type (wt) or *pp5^Δ^/Df* females were tested by crossing to wild type (first and second bars, p<0.01) or *pp5^Δ^/Df* males (third bar, p<0.001). We found a significant decrease in progeny number when *pp5^Δ^/Df* siblings were crossed, suggesting the role of PP5 in the early development of fruit flies. Significance was determined by Welch’s test.

## Data Availability

Not applicable.

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
