# Peer review of "Plk4 Is a Novel Substrate of Protein Phosphatase 5"

_ijms, 2023, doi:10.3390/ijms24032033_

Round 1

Reviewer 1 Report

This is an excellent study that pursues a physical interaction between PP5 and Plk4 in Drosophila.  The authors carry out some meticulous biochemical studies that convincingly show that PP5 binds to the kinase domain of Plk1 and that the kinase id dephosphorylated by PP5. However, this dephosphorylation has no affect upon the ability of Plk4 to phosphorylate its main substrate in centriole duplication, Ana1, or to prevent the interaction between Ana1 and Sas6. They show that PP5 does localise with centrosomes, partially purified on a sucrose gradient and interestingly, the centriolar proteins Sas4, Cep135 and Cp110 bind preferentially to dephosphorylated Plk4 but the significance of this is unclear.  The team generate PP5 mutants but these appear to have no centriolar defects although both males and females show reduced fertility.  This is a solid body of work that eliminates the obvious possibilities for the functional importance of the association between PP5 and Plk4 and, whereas further studies would be required to establish a role for PP5 at the centrosome, it is important to publish the current study to facilitate future work.  I recommend publication of this excellent study without the need for any revision.

Author Response

Please see the attached file for details:  Response to Reviewers’ comments.pdf

Answer:

We thank Reviewer 1 for the supporting comments. We are glad that she/he considers the results we presented worthy of publication.

We also found interesting the preferential binding of centrosomal proteins to phosphorylated or dephosphorylated Plk4. Therefore, we repeated the main experiment presented in Figure 5. We purified new p-MBP-Plk4-TE alone or from bacteria in which we co-expressed with His6-PP5 or its inactive form, respectively, and repeated the binding with Asterless, the main partner of Plk4 in fruit flies. This time we included extra negative and positive controls and found that Asterless also shows preference to dephosphorylated Plk4 (similarly to Sas4, Cep135 and Cp110), however, it also binds weakly to the phosphorylated kinase, in vitro. This was repeated twice with the same result, therefore we decided to replace Figure 5 to a more informative and new image and we also changed the text in the manuscript as well as supplementary information (Figure S4) accordingly. 

Reviewer 2 Report

It is worth being accepted.

Author Response

Please see the attached file for details:  Response to Reviewers’ comments.pdf

Reviewer 3 Report

In this paper the authors tested hypothesis that PP5 docks at Plk4, based on the information of previous reports.  The interaction was tested using biochemical assays with recombinant proteins.  Also the functions of PP5 in regulating Plk4 was examined using the cell culture and fly.  Biochemical assays, such as the pulldown with 35S-labeled proteins, show the binding of PP5 to Plk4 and structural elements involved in the interaction.  Phosphatase assay was assessed based on the gelshift of phospho-proteins.  The PP5-Plk4 complex formation in cells was confirmed using sucrose gradient assay and BiFC assay.  Based on the data, the authors claimed that Plk4 function is regulated through the interaction with PP5.  The interaction between PP5 and Plk4, which is somewhat subtle, is clearly evidenced by the data of thoroughly conducted binding assays.  However, to support the conclusion, the biological significance of the interaction, quantitive analyses for the presented data are needed.

In Figure 1 panel b, interactions of Plk4-ND with GFP-PP5(WT) and H304Q form are compared.  Indeed, the H304Q form was co-precipitated with anti-FLAG beads.  However, it is not clear if Plk4-ND binds WT protein as well as H304Q, because the extent of FLAG-Plk4 on the beads is much higher in the lane of H304Q.  In addition, this experiment needs a control, in which the beads without FLAG-Plk4 are mixed with GFP, PP5 WT and H304Q.

In Figure 2 panel (b), the authors concluded the kinase domain is the docking site of PP5.  Compared with the full-length Plk4, only faint band of the bound delta-KD form is shown.  However, the extent of the input delta-KD is much less, compared with the full-length Plk4.  On the other hand, bound proteins of neither delta-PB1-2, CC-DRE or PB1+PB2 were invisible.  Quantitive analyses to compare the ratio of bound protein to input are needed for supporting the conclusion.

Figure 3 shows dephophorylation of Plk4 by recombinant PP5.  In this assay, excessive amount of PP5 seems to be added to Plk4.  The ration between PP5 and Plk4 does not seem to represent physiological dephosphorylation.  

In Figure 4 panel (a), the extent of P-GST-Ana2 seems to be lower in the lane of H326N form, compared with PP5 WT.  It is not clear whether the quantitive argument of the Ana2 phosphorylation is possible from the data.  Also, does PP5 possibly dephosphorylate Ana2?

In the data of Figure 6 majority of PP5 dominantly exists in the centrosomal fraction, compared with the cytoplasm.  Are there any previous reports supporting the imaging data, such as immunocytochemistry or GFP-PP5 imaging?

In Figure 7 there is no description of the image analysis of the YFP-complementation assay.  How are thresholds of imaging defined? How many cells are counted? Quantitative analysis is needed for this innovative assay.

Figure 8 shows that  PP5 gene ablation suppressed fertility without changing mitotic index.  On the other hand, the authors claimed that PP5 binds to Plk4 and regulates centrosomal functions based on the other data.  Therefore, the phenotype of PP5 gene ablation seems to be irrelevant to the PP5-Plk4 interaction, the main focus of this paper.  This gap in the findings should be clearly noted and the possible mechanisms should be interpreted in text.  

Author Response

(The authors gave the same response as above.)

Round 2

Reviewer 3 Report

The authors provided point-by-point response and revised the paper throughly.  To address the concerns, new experiments with more proper controls were added and the text was revised based on the new data.  Although quantitative analyses that were requested were not provided in this revision, the authors re-sentenced the arguments and conclusion to fit to the current form of the data.  In addition, the revision of sentences made this paper more understandable.